Establishment and validation of a heart failure risk prediction model for elderly patients after coronary rotational atherectomy based on machine learning

Zhang Lixiang
Zhou Xiaojuan
Cao Jiaoyu caojiaoyu@126.com
Department of Cardiology, The First Affiliated Hospital of USTC, Division of Life Science and Medicine, University of Science and Technology of China , Hefei, Anhui , China
Adegboye Oyelola
Electronic publication date: 2024 Jan 31
Publication date: 2024
Volume: 12
Electronic Location ID: e16867
Received 2023 Jun 26; Accepted 2024 Jan 10
Copyright: © 2024 Zhang et al.
Copyright year: 2024
Copyright holder: Zhang et al.
License: This is an open access article distributed under the terms of the Creative Commons Attribution License, which permits unrestricted use, distribution, reproduction and adaptation in any medium and for any purpose provided that it is properly attributed. For attribution, the original author(s), title, publication source (PeerJ) and either DOI or URL of the article must be cited.
License URL: https://creativecommons.org/licenses/by/4.0/

Keywords: Coronary rotational atherectomy, Heart failure, Risk, Machine learning, Prediction model

Funding: China University of Science and Technology 2022xjyxm089 This study was funded by the University-level undergraduate quality engineering project of China University of Science and Technology in 2022 (No.:2022xjyxm089). The funders had no role in study design, data collection and analysis, decision to publish, or preparation of the manuscript.

==============================
Objective

To develop and validate a heart failure risk prediction model for elderly patients after coronary rotational atherectomy based on machine learning methods.

Methods

A retrospective cohort study was conducted to select 303 elderly patients with severe coronary calcification as the study subjects. According to the occurrence of postoperative heart failure, the study subjects were divided into the heart failure group (n = 53) and the non-heart failure group (n = 250). Retrospective collection of clinical data from the study subjects during hospitalization. After processing the missing values in the original data and addressing sample imbalance using Adaptive Synthetic Sampling (ADASYN) method, the final dataset consists of 502 samples: 250 negative samples (i.e., patients not suffering from heart failure) and 252 positive samples (i.e., patients with heart failure). According to a 7:3 ratio, the datasets of 502 patients were randomly divided into a training set (n = 351) and a validation set (n = 151). On the training set, logistic regression (LR), extreme gradient boosting (XGBoost), support vector machine (SVM), and lightweight gradient boosting machine (LightGBM) algorithms were used to construct heart failure risk prediction models; Evaluate model performance on the validation set by calculating the area under the receiver operating characteristic curve (ROC) curve (AUC), sensitivity, specificity, positive predictive value, negative predictive value, F1-score, and prediction accuracy.

Result

A total of 17.49% of 303 patients occured postoperative heart failure. The AUC of LR, XGBoost, SVM, and LightGBM models in the training set were 0.872, 1.000, 0.699, and 1.000, respectively. After 10 fold cross validation, the AUC was 0.863, 0.972, 0.696, and 0.963 in the training set, respectively. Among them, XGBoost had the highest AUC and better predictive performance, while SVM models had the worst performance. The XGBoost model also showed good predictive performance in the validation set (AUC = 0.972, 95% CI [0.951–0.994]). The Shapley additive explanation (SHAP) method suggested that the six characteristic variables of blood cholesterol, serum creatinine, fasting blood glucose, age, triglyceride and NT-proBNP were important positive factors for the occurrence of heart failure, and LVEF was important negative factors for the occurrence of heart failure.

Conclusion

The seven characteristic variables of blood cholesterol, blood creatinine, fasting blood glucose, NT-proBNP, age, triglyceride and LVEF are all important factors affecting the occurrence of heart failure. The prediction model of heart failure risk for elderly patients after CRA based on the XGBoost algorithm is superior to SVM, LightGBM and the traditional LR model. This model could be used to assist clinical decision-making and improve the adverse outcomes of patients after CRA.

Introduction

Coronary artery calcification (CAC) is the sclerosis caused by calcium salt deposition in the coronary artery, which is common in patients with coronary heart disease, with a high incidence and case fatality rate (Junwei & Wei, 2021; Rui et al., 2023). For patients with severe calcification, percutaneous coronary intervention (PCI) treatment may face significant challenges, such as vascular detachment, obstructed stent delivery, and insufficient stent opening (Xiaonan et al., 2021). In addition, coronary calcification can also lead to myocardial perfusion damage and polymer coating failure on drug eluting stents (Guijun & Qi, 2019). Therefore, timely detection and management of coronary artery calcification should be carried out before PCI surgery to avoid the occurrence of adverse cardiac injury events.

Coronary rotational atherectomy (CRA) is a method to treat severe coronary artery calcification. The calcified tissue is grinded into tiny particles by using a high-speed rotating rotary head, so that the human Phagocyte system can engulf (Qian, Tao & Jun, 2020; Uetani & Amano, 2018). With the popularization of PCI surgery in China, the age of patients participating in interventional therapy is generally older, and the number of patients with severe coronary artery calcification symptoms has significantly increased. Therefore, the application of CRA for vascular pretreatment has become very important (Jichun et al., 2020). Research has found that elderly patients often suffer from multiple underlying diseases and decreased cardiovascular function, which increases the risk of adverse cardiac events after CRA. Among them, heart failure is one of the common adverse events and one of the main causes of postoperative death for patients (Zhengwei et al., 2023). Therefore, early prediction of heart failure after coronary rotational atherectomy is of great significance for the prognosis of patients.

With the improvement of computer performance and optimization of algorithms, machine learning has been widely applied in clinical research. Over the past decade, many research teams have attempted to create different machine learning models for predicting the risk of clinical related events (Lauritsen et al., 2021; Cilloniz et al., 2023). However, there is currently no progress in machine learning related research on predicting the risk of heart failure in patients after CRA. Therefore, this study will take patients after CRA as the research object, use the three most widely used Ensemble learning algorithms to establish the prediction model of patients’ postoperative heart failure risk, and compare it with the traditional logistic regression model, in order to provide new ideas for clinicians to identify high-risk patients with heart failure early and carry out precise intervention.

Materials and Methods

Research subject

This study is a retrospective cohort study, in which 303 elderly patients with severe coronary calcification were selected consecutively as the study subjects, and the time span is from June 2017 to June 2021. These patients were hospitalized in the Department of Cardiology of a tertiary hospital in Anhui Province, China and underwent CRA. Inclusion criteria of research subjects: (1) Over 60 years old. (2) All patients met the indications for CRA treatment, patient or family members have signed the informed consent form for the surgery. (3) Diagnosed with coronary intimal calcification at grade IV, with strong echoic masses along the vessel wall and a maximum calcification arc of >270°, as confirmed by intravascular ultrasound examination. Exclusion criteria of research subjects: (1) Presence of ulcerative or thrombotic lesions that can be exacerbated by rotational atherectomy. (2) Presence of chronic occlusive lesions. (3) Severe mental illness and inability to communicate normally. (4) Lesions that are prone to thrombosis and embolism, such as degenerative saphenous vein bridge lesions. (5) Presence of intimal tearing lesions that can be worsened by rotational atherectomy. (6) Severe angulation (>60°) lesions. (7) Acute myocardial infarction in the acute phase (≤7 days). (8) Severe functional impairment of vital organs such as liver, lungs, kidneys, etc. The standard for coronary artery calcification lesions is that after completing coronary angiography, the stenosis of blood vessel diameter exceeds 50% and the reference diameter is greater than or equal to 1.5 millimeters (Zhengwei et al., 2023).

Data collection

Retrospective collection of postoperative clinical data of the research subjects through the Hospital information system (HIS). The data included gender, age, complications (chronic renal insufficiency, ischemic cardiomyopathy, cerebrovascular disease, hypertension, diabetes), left ventricular ejection fraction (LVEF), N-terminal pro brain natriuretic peptide (NT-proBNP), serum creatinine, fasting blood glucose, Glycated hemoglobin, preoperative creatine kinase isoenzymes (CK-MB), preoperative troponin, hemoglobin, total cholesterol (TC), triglyceride (TG), low density lipoprotein cholesterol (LDL-C) and very low-density lipoprotein cholesterol (VLDL-C), 19 indicators in total. Calculated the incidence of postoperative heart failure in subjects, refer to the diagnostic criteria in the “Chinese Guidelines for the Diagnosis and Treatment of Heart Failure 2018”, Jianjun (2019) and diagnosed heart failure patients through NYHA grading, cardiac ultrasound, and other examinations. According to the occurrence of postoperative heart failure, the study subjects were divided into two groups, with 53 cases in the heart failure group and 250 cases in the non-heart failure group. This study was approved by the Medical Research Ethics Committee of the First Affiliated Hospital of University of Science and Technology of China (approval number: 2021-RE-026). Due to the retrospective nature of the study, the informed consent of the study object was exempted.

Data cleaning and pre-processing

Missing data can make analysis more difficult and may lead to deviations in the analysis results, thereby reducing accuracy (Yun et al., 2023). In order to deal with missing data, Random forest, an integrated classifier based on decision tree, can be used. It has strong anti noise ability, is not easily affected by outliers, and does not limit the type of data distribution (Xiaoqin & Yuying, 2017). If the missing value of a variable exceeds 20%, it needs to be excluded from the final dataset (Zhang et al., 2023). In this study, the random forest regression method was used to impute data for indicators with the percentage of missing data less than 20%, and the median was used to replace the outlier in the dataset. In addition, all continuous variables in our study have undergone Z-score normalization so as to scale each continuous variable to a distribution with mean 0 and standard deviation 1, while categorical variables have been processed with one-hot encoding. The purpose of one-hot encoding for categorical variables is to convert non-numeric category data into a numerical form that machine learning models can understand. Meanwhile, the purpose of Z-score normalization for continuous variables is to eliminate the dimensional inconsistencies and centralize the data, ensuring that the scales of different features are consistent, thereby enhancing the efficiency and accuracy of model training. In binary classification tasks, when the ratio of negative to positive samples in the dataset approaches 1:1, it can effectively avoid bias introduced by the samples. Therefore, to enhance the ultimate predictive performance of the model, this study employs adaptive synthetic sampling (ADASYN) technology to mitigate the issue of class imbalance. Compared to traditional random oversampling methods, ADASYN not only achieves a balance between negative and positive samples but also reduces the occurrence of overfitting (He et al., 2008). After processing the missing values in the original data and addressing sample imbalance, the final dataset consists of 502 samples: 250 negative samples (i.e., patients not suffering from heart failure) and 252 positive samples (i.e., patients with heart failure).

Model construction

In order to improve the generalization ability of the model and reduce the occurrence of overfitting, this study referred to relevant research (Wang et al., 2021) and used the LASSO regression analysis to screen for meaningful variables, which were included in the model construction. The dataset of 502 study subjects was randomly divided into a training set (351 samples) and a testing set (151 samples) using a 7:3 ratio. This randomization was achieved through the combined application of various modules within SPSS 25.0, including the Random Number Generation, Compute Variable, Rank Cases, and Select Cases modules, and four algorithms were used to construct a heart failure risk prediction model, namely the traditional logistic regression model (LR), extreme gradient boosting model (XGBoost), Support vector machine (SVM) and lightweight gradient boosting machine (LightGBM). The last three algorithms are the most widely used Ensemble learning models, and they can combine multiple learners to obtain better generalization performance. The testing set is used to validate and evaluate the performance of the selected model.

Model evaluation

Evaluate the predictive performance of various machine learning models using multiple evaluation indicators, including area under the receiver operating characteristic curve(ROC) curve (AUC), sensitivity, specificity, positive predictive value, negative predictive value, prediction accuracy, and F1-score (a weighted average of accuracy and recall) (Xin et al., 2023). At the same time, the clinical applicability of the model was evaluated using the decision curve analysis (DCA) curve, Jingjing et al. (2023) the calibration degree of the model is evaluated by the calibration curve and the interpretability of the model was evaluated using the Shapley additive explanation (SHAP) method (Ruihao et al., 2022).

Statistical analysis

Statistical software SPSS 25.0, R software (3.6.3; R Core Team, 2020), and Python software (3.7.0) were used for data analysis. M (P25, P75) is used to represent econometric data with skewed distribution, and Mann Whitney U test is used for inter group comparison. The frequency (%) was used to represent the counting data, and Pearson Chi-squared test was used to compare between groups. All test results were obtained from bilateral tests, and when P < 0.05, the difference was statistically significant.

Results

Comparison of clinical data of patients

Among the 303 study subjects, 53 patients occured postoperative heart failure, with a heart failure incidence rate of 17.49%. Compared with the non-heart failure group, the heart failure group has a higher proportion of chronic renal insufficiency, a lower LVEF, and higher levels of NT-proBNP, serum creatinine, fasting blood glucose, blood cholesterol, triglyceride, and low-density lipoprotein cholesterol, with statistically significant differences (all P < 0.05), as shown in Table 1.

Table 1 Comparison of clinical data between two groups of patients.

Variables	Total (n = 303)	Non-heart failure group (n = 250)	Heart failure group (n = 53)	χ2/z	P	
Gender, n (%)				1.188a	0.276	
Male	174 (57.43)	140 (56.00)	34 (64.15)			
Female	129 (42.57)	110 (44.00)	19 (35.85)			
Combined with chronic renal insufficiency, n (%)	14 (4.62)	8 (3.20)	6 (11.32)	6.544a	0.011	
Combined with ischemic cardiomyopathy, n (%)	17 (5.61)	12 (4.80)	5 (9.43)	1.773a	0.183	
Combined with cerebrovascular disease, n (%)	60 (19.80)	51 (20.40)	9 (16.98)	0.322a	0.570	
Combined with diabetes, n (%)	99 (32.67)	81 (32.40)	18 (33.96)	0.049a	0.826	
Concomitant with hypertension, n (%)	173 (57.10)	140 (56.00)	33 (62.26)	0.700a	0.403	
Age	72.00 (67.00– 77.00)	72.00 (67.00–76.00)	72.00 (67.00– 77.00)	−0.917b	0.359	
LVEF	57.00 (47.00–65.00)	58.00 (48.00–65.00)	48.00 (35.00–61.00)	3.619b	<0.001	
Preoperative CK-MB	16.00 (12.00–24.00)	15.00 (12.00–24.00)	17.00 (14.00– 23.00)	−1.867b	0.062	
Preoperative troponin	0.12 (0.01–0.65)	0.13 (0.01–0.65)	0.08 (0.01–0.64)	−0.280b	0.773	
NT-proBNP	1,442.00 (408.00–2,856.00)	1,223.00 (290.00–2,533.00)	3,471.00 (1,897.00–6,987.00)	−5.909b	<0.001	
Serum creatinine	74.00 (62.00–93.00)	73.00 (60.00– 90.00)	90.00 (70.00–136.00)	−4.158b	<0.001	
Fasting blood glucose	5.51 (4.79–7.17)	5.40 (4.75–6.85)	7.10 (5.36–9.13)	−3.984b	<0.001	
HbA1c	6.80 (5.90–7.90)	6.80 (5.90–7.79)	7.00 (5.90– 8.68)	−1.245b	0.213	
hemoglobin	121.00 (110.00–131.00)	122.00 (110.00–132.00)	112.00 (108.00–130.00)	1.576b	0.115	
Blood cholesterol	3.73 (3.10–4.49)	3.65 (3.08–4.37)	4.36 (3.27– 6.21)	−3.127b	0.002	
Triglyceride	1.22 (0.95–1.61)	1.17 (0.95–1.57)	1.39 (1.12–2.42)	−2.410b	0.016	
Low density lipoprotein cholesterol	1.90 (1.49–2.44)	1.87 (1.48–2.33)	2.13 (1.58– 2.68)	−-2.390b	0.017	
Very low-density lipoprotein cholesterol	0.84 (0.68–1.02)	0.83 (0.66–1.01)	0.88 (0.76–1.06)	−1.827b	0.068	
Notes:

a Pearson Chi-squared test.

b Mann Whitney rank sum test.

LVEF, left ventricular ejection fraction; NT-proBNP, N-terminal pro brain natriuretic peptide; CK-MB, creatine kinase isoenzymes; HbA1c, Glycated hemoglobin.

Comparison of training set and testing set

This study randomly divided the data of 502 patients into a training set and a testing set in a 7:3 ratio, consisting of 351 and 151 patients, respectively. There was no statistically significant difference in clinical data between the two groups of patients (P > 0.05), indicating that the two datasets were homogeneous and comparable (P > 0.05), as shown in Table 2.

Table 2 Comparison of training set and testing set.

Variables	Category	Total (n = 502)	Training set (n = 351)	Testing set (n = 151)	χ2/z	P	
Heart failure, n (%)	No	250 (49.80)	176 (50.14)	74 (49.01)	0.054a	0.815	
	Yes	252 (50.20)	175 (49.86)	77 (50.99)			
Combined with chronic renal insufficiency, n (%)	No	485 (96.61)	337 (96.01)	148 (98.01)	1.293a	0.255	
	Yes	17 (3.39)	14 (3.99)	3 (1.99)			
Combined with Ischemic cardiomyopathy, n (%)	No	481 (95.82)	337 (96.01)	144 (95.36)	0.110a	0.740	
	Yes	21 (4.18)	14 (3.99)	7 (4.64)			
Combined with cerebrovascular disease, n (%)	No	435 (86.65)	307 (87.46)	128 (84.77)	0.664a	0.415	
	Yes	67 (13.35)	44 (12.54)	23 (15.23)			
Combined with diabetes, n (%)	No	359 (71.51)	254 (72.36)	105 (69.54)	0.415a	0.520	
	Yes	143 (28.49)	97 (27.64)	46 (30.46)			
Combined with hypertension, n (%)	No	245 (48.80)	172 (49.00)	73 (48.34)	0.018a	0.892	
	Yes	257 (51.20)	179 (51.00)	78 (51.66)			
Gender, n (%)	Male	340 (67.73)	244 (69.52)	96 (63.58)	1.704a	0.192	
	Female	162 (32.27)	107 (30.48)	55 (36.42)			
Age	/	72.00 (67.00, 76.00)	72.00 (67.00, 76.00)	72.00 (67.00, 76.00)	−0.798b	0.424	
LVEF	/	54.00 (44.00, 62.00)	55.00 (43.00, 62.00)	53.00 (45.00, 63.00)	−0.023b	0.982	
Preoperative CK-MB	/	16.12 (13.00, 24.00)	16.57 (13.00, 24.88)	16.01 (13.00, 23.93)	0.218b	0.827	
Preoperative troponin	/	0.14 (0.01, 0.65)	0.10 (0.01, 0.64)	0.23 (0.01, 0.65)	−1.317b	0.181	
NT-proBNP	/	1880.00 (643.00, 3471.00)	1858.00 (676.00, 3499.00)	1897.00 (641.00, 2985.00)	0.506b	0.613	
Serum creatinine	/	82.00 (66.66, 105.00)	81.79 (68.04, 105.00)	82.00 (64.00, 105.00)	0.339b	0.734	
Fasting blood glucose	/	6.62 (5.04, 7.84)	6.54 (5.05, 7.78)	6.76 (5.05, 8.01)	−0.886b	0.376	
HbA1c	/	6.98 (5.92, 8.10)	6.98 (5.90, 8.05)	7.00 (6.03, 8.16)	−0.704b	0.481	
Hemoglobin	/	119.83 (109.57, 130.00)	120.06 (109.42, 131.00)	116.00 (110.00, 127.00)	1.665b	0.096	
Blood cholesterol	/	4.05 (3.25, 4.99)	4.12 (3.25, 5.09)	3.90 (3.25, 4.71)	1.129b	0.259	
Triglyceride	/	1.31 (1.06, 1.76)	1.35 (1.06, 1.75)	1.24 (1.06, 1.85)	0.035b	0.973	
Low density lipoprotein cholesterol	/	2.02 (1.53, 2.59)	2.02 (1.57, 2.63)	2.03 (1.52, 2.48)	0.640b	0.523	
Very low-density lipoprotein cholesterol	/	0.87 (0.74, 1.04)	0.88 (0.75, 1.04)	0.87 (0.74, 1.04)	0.540b	0.590	
Notes:

a Pearson Chi-squared test.

b Mann Whitney rank sum test.

LVEF, left ventricular ejection fraction; NT-proBNP, N-terminal pro brain natriuretic peptide; CK-MB, creatine kinase isoenzymes; HbA1c, Glycated hemoglobin.

Feature variable screening results

In the training set, the LASSO regression was used to screen the characteristic variables of 19 indicators, and the variable of non-zero regression coefficient corresponding to the Lambda coefficient of the minimum distance standard error was selected as the characteristic variable through 10 times cross validation. The LASSO regression results show that the Lambda coefficient of the minimum distance standard error (Lambda.1se) is 0.037, and the corresponding characteristic variables include triglyceride, age, blood cholesterol, fasting blood glucose, serum creatinine, NT-proBNP and LVEF; the Lambda coefficient of minimum mean square error (Lambda.min) is 0.014, and the corresponding characteristic variables include VLDL-C, triglyceride, blood cholesterol, HbA1c, fasting blood glucose, creatinine, NT-proBNP, LVEF, chronic renal insufficiency, ischemic cardiomyopathy, cerebrovascular disease, diabetes, age and gender (Fig. 1).

Figure 1 Lasso regression analysis results.

(A) Lasso regression coefficient diagram; (B) lasso regression cross validation statistical chart. The two vertical dashed lines in the chart represent the logarithmic Lambda coefficient of the minimum mean square error (dashed line on the left) and the logarithmic Lambda coefficient of the standard error of the minimum distance (dashed line on the right).

Model construction

In the training set, four algorithms including LR, XGboost, LightGBM and SVM, were used to build a machine learning prediction model including seven variables (triglyceride, blood cholesterol, fasting blood glucose, creatinine, NT-proBNP, LVEF, age) based on the corresponding characteristic variables of Lambda.1se from the LASSO regression, and 10 times cross validation was used to internally verify the built prediction model. The results showed that the XGBoost model had the highest AUC in the training set and the internal validation set, suggesting that XGBoost was the optimal model, as shown in Table 3 and Fig. 2.

Table 3 Comparison results of multiple models.

Classification model	AUC	Cutoff	Accuracy	Sensitivity	Specificity	Positive predictive value	Negative predictive value	F1-score	
Training set									
XGBoost	1.000	0.831	0.997	1.000	1.000	1.000	0.994	1.000	
Logistic	0.872	0.475	0.808	0.845	0.778	0.790	0.830	0.817	
LightGBM	1.000	0.890	0.997	1.000	1.000	1.000	0.994	1.000	
SVM	0.699	0.463	0.686	0.690	0.690	0.686	0.686	0.688	
Internal validation set									
XGBoost	0.972	0.831	0.889	0.978	0.909	0.929	0.864	0.951	
Logistic	0.863	0.475	0.801	0.813	0.841	0.785	0.825	0.796	
LightGBM	0.963	0.890	0.863	0.977	0.915	0.940	0.823	0.957	
SVM	0.696	0.463	0.678	0.733	0.689	0.684	0.682	0.698	

Figure 2 ROC curve of multiple models.

(A) ROC curve in training set; (B) ROC curve in the internal validation set.

Model performance evaluation

Testing the predictive ability of the XGBoost algorithm on the risk of heart failure in patients after CRA in the testing set. The results showed that the AUC of the XGBoost model in the testing set was 0.972, the prediction accuracy was 0.921, the sensitivity was 1.000, the specificity was 0.892, the positive predictive value was 0.911, the negative predictive value was 0.931, and the F1-score was 0.954. See Fig. 3A for the receiver operating characteristic of XGBoost model in the testing set. The clinical applicability of the XGBoost model was evaluated using the DCA curve in the testing set. The results of the DCA curve showed that when the threshold probability of heart failure in patients was between 1–90%, the net benefit of using the XGBoost model for risk assessment was the highest, significantly better than the “full intervention” and “no intervention” schemes. This suggests that the XGBoost model has good clinical applicability, as shown in Fig. 3B. The calibration curve in the testing set suggested a good agreement between the predicted probability of the XGBoost model and the frequency of postoperative heart failure, suggesting that the XGBoost model is well calibrated, as detailed in Fig. 3C.

Figure 3 XGBoost’s ROC curve, DCA curve and calibration curve in the testing set.

(A) ROC curve; (B) DCA curve; (C) calibration curve.

Interpretability analysis of the model

In order to deeply explore the main influencing factors of heart failure occurrence and improve the interpretability of classification models, this study used the SHAP method to conduct interpretability analysis on the XGBoost model (Jiasi, 2023). Figure 4A showed the importance ranking of each feature variable in the XGBoost model. It can be seen from the figure that the order of importance is fasting blood glucose, NT-proBNP, blood cholesterol, serum creatinine, triglyceride, LVEF and age, and the importance of these seven characteristic variables is relatively concentrated, which can be regarded as important factors affecting the occurrence of heart failure. Figure 4B showed the distribution of SHAP values for each feature variable, sorted by the importance of each feature from top to bottom. The horizontal axis represents the SHAP value of the model, and the color of the points represents the size of the feature values. Red indicates a large feature value, while blue indicates a small feature value. A positive SHAP value indicates a positive contribution to the model’s prediction of heart failure, while a negative SHAP value indicates a negative contribution to the model’s prediction of heart failure. It can be seen from Fig. 4B that fasting blood glucose has the greatest impact on the prediction results of the model, and with the increase of fasting blood glucose value, the probability of heart failure predicted by the sample will increase, that is, this feature has a positive impact on the prediction of heart failure, and the trend of NT-proBNP, blood cholesterol, serum creatinine, triglyceride and age is similar. The trend of LVEF is opposite to that of blood cholesterol. As the LVEF value increases, the probability of the sample being diagnosed with heart failure decreases.

Figure 4 Statistical plots of the SHAP analysis.

(A) Order plot of variable importance for SHAP analysis; (B) statistical graph of variable contribution in SHAP analysis.

The SHAP method can not only analyze the overall influencing factors of the prediction model, but also analyze individual influencing factors (Ruihao et al., 2022). For example, we provide two typical examples to illustrate the interpretability of the model. One postoperative patient without heart failure had a low SHAP prediction score (0.00), as shown in Fig. 5A; while the other postoperative patient who developed heart failure had a high SHAP score (0.98), as shown in Fig. 5B.

Figure 5 Example of SHAP interpretation in patients with heart failure.

(A) Individual efforts by patients without heart failure; (B) individual efforts by patients with heart failure.

Discussion

Research has shown that heart failure is closely related to coronary artery disease, and heart failure can exacerbate coronary artery disease (Heart Failure Group of Cardiology Branch of Chinese Medical Association, 2018). Therefore, it is necessary to assess the risk of heart failure as early as possible for patients after CRA and take corresponding preventive measures for those at high risk of heart failure, in order to prevent the deterioration of the condition and save the patient’s life. However, the commonly used scoring tools currently require manual operation, which is time-consuming, labor-intensive, and inefficient. Accurate, convenient, and fast disease assessment can assist clinical decision-making, timely take treatment measures, and is of great significance for the risk assessment of heart failure and the prognosis of patients.

With the continuous development of algorithms and computer hardware, and the arrival of the era of Big data, machine learning has shown great advantages in mining and processing medical data. At present, it has been widely used in predicting the occurrence and prognosis of clinical diseases (Jie, 2022; Peng et al., 2022) such as predicting the risk of coronary heart disease (Jialun et al., 2023; Haoxuan et al., 2022) the risk of recurrence after radiofrequency ablation of atrial fibrillation (Huanxu et al., 2022) and adverse outcomes after acute coronary syndrome (D’Ascenzo et al., 2021).

This study used LR, XGBoost, SVM, and LightGBM algorithms to establish risk prediction models for heart failure in elderly patients after CRA, and compared the predictive capabilities of these four models. At present, XGBoost, SVM and LightGBM are the three most widely used Ensemble learning algorithms. Among them, LightGBM and XGBoost mainly use the Boosting method, while SVM uses the nonlinear mapping theory to find the optimal plane to partition the feature space, fully consider the dependency between attributes, and add association arcs to expand the structure of the naive Bayesian model, thus significantly improving the classification effect (Yingying et al., 2022). LR is a classic regression analysis method commonly used to study disease risk factors and predict the probability of disease occurrence. However, Xiao et al. (2019) found in a study that compared to some machine learning algorithms, traditional LR has greater prediction errors and poorer prediction performance. The results of this study confirmed this conclusion, that is the AUC of the LR model in the training set and internal validation set are 0.891 and 0.865, respectively, and the accuracy is 0.790 and 0.760, both lower than the XGBoost model. The XGBoost model has better discrimination and can more accurately predict the risk of heart failure, demonstrating excellent performance in identifying patients at high risk of heart failure.

In addition, the univariate analysis results of this study found that compared with the non-heart failure group, the number of patients with chronic renal insufficiency was higher, the LVEF was lower, and the levels of NT-proBNP, serum creatinine, fasting blood glucose, blood cholesterol, triglycerides, and low-density lipoprotein cholesterol were higher, with statistically significant differences (all P < 0.05). This conclusion is consistent with the LASSO regression screening of six characteristic variables: blood cholesterol, triglycerides, serum creatinine, fasting blood glucose, NT-proBNP and LVEF, and with the SHAP method suggesting that blood cholesterol, serum creatinine, triglycerides, fasting blood glucose, NT-proBNP and LVEF are all important factors affecting the occurrence of heart failure. In addition, LASSO regression also suggests that age is one of the characteristic variables predicting the occurrence of heart failure after CRA, and the SHAP method also suggests that age is one of the important predictors of the occurrence of heart failure after CRA. Previous research findings are also similar with the results of this study, that is, LVEF < 45% and NT-proBNP ≥ 1,800 ng/L at admission are independent risk factors for heart failure after percutaneous coronary intervention (PCI) in patients with acute myocardial infarction (AMI) (Chenglong et al., 2022). High fasting blood glucose is a high risk factor for elderly hypertensive patients with heart failure (Jianfeng & Xiaoyan, 2017). Serum creatinine is an independent influencing factor for the long-term prognosis of chronic heart failure, and has an overall adverse effect on the long-term mortality rate of chronic heart failure (Houliang et al., 2022). Abnormal serum cholesterol levels may increase the risk of heart failure (Guoying, 2010). According to the study on the relationship between lipid levels and heart function in patients with chronic heart failure conducted by Hui & Xiaofang (2005) triglyceride is independently correlated with the occurrence and severity of heart failure, and triglyceride can be used as a reference indicator to determine the severity of heart failure, which is consistent with the results of this study, which found that triglyceride is an important predictor of the occurrence of heart failure in patients after CRA. Meili & Chunge (2023) found that age over 65 years old is one of the important factors affecting the occurrence of heart failure in patients with acute myocardial infarction during hospitalization, which is consistent with the finding in this study that age is an important predictor of the occurrence of heart failure in patients after CRA, suggesting that targeted and reasonable interventions should be carried out for elderly patients to reduce the incidence of heart failure.

This study also has some limitations. Firstly, it was a single center retrospective study conducted at a tertiary hospital in Anhui Province, China. Although the researchers have tried their best to retrospectively collect the clinical data of heart failure and non-heart failure after CRA in the elderly, the sample size is still smaller than some large studies, so more multicenter, large sample and prospective cohort study are needed to evaluate the performance of the XGBoost model. Secondly, this study wasn’t validated by external datasets, and the promotion of the research results may be limited to some extent. Therefore, it is difficult to ensure the universality and promotion ability of the research results in other regions. Conducting multi-center validation research is particularly important for evaluating the model’s generalization ability. Finally, our study is a retrospective, single-center study, and the limitations of the indicators collected retrospectively have resulted in the exclusion of specific factors related to coronary rotational atherectomy, such as lesion location (proximal, midshaft, or distal), maximum burr diameter, number of burrs used, vessel characteristics (presence of bifurcations, tortuosity, or complete occlusion), and target vessel involvement. These omissions may restrict the applicability of our findings to certain populations. Therefore, it is particularly necessary to conduct a multi-center, large-scale, prospective study that collects a broader array of research indicators. Such a study would not only allow us to assess the generalization ability and robustness of the predictive model constructed in our research but also to optimize the specificity and targeting of post-operative heart failure prediction models by including more specific indicators.

Conclusion

It is feasible to use four Ensemble learning algorithms LR, XGBoost, SVM and LightGBM to establish the risk prediction models of heart failure for patients after CRA. The XGBoost model has the best predictive performance. This model can identify high-risk postoperative patients with heart failure in advance, help medical staff make treatment decisions and adjust treatment plans, thereby reducing the occurrence of adverse outcomes. This has important clinical application significance for clinical medical workers.

Supplemental Information

Supplemental Information 1 Raw data.

Click here for additional data file.

Supplemental Information 2 Code book for the raw data.

Click here for additional data file.

Supplemental Information 3 Results of Sensitivity Analysis.

Click here for additional data file.

Additional Information and Declarations

Competing Interests

Author Contributions

Data Availability

The authors declare that they have no competing interests.

Lixiang Zhang conceived and designed the experiments, performed the experiments, analyzed the data, prepared figures and/or tables, and approved the final draft.

Xiaojuan Zhou performed the experiments, analyzed the data, prepared figures and/or tables, authored or reviewed drafts of the article, and approved the final draft.

Jiaoyu Cao conceived and designed the experiments, authored or reviewed drafts of the article, and approved the final draft.

The following information was supplied regarding data availability:

The raw data are available in the Supplemental Files.

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
