# Peer review of "Establishment and validation of a heart failure risk prediction model for elderly patients after coronary rotational atherectomy based on machine learning"

_PeerJ, doi:10.7717/peerj.16867_

## Round 0.1 · original submission · Major Revisions

Line 61: What is PCI?
Lines 115-122: Did you conduct sensitivity analyses on the imputed vs original data? If not, please do.
Present in the supplementary material detailed information on the missing % in this study.

Reviewer 1 ·

Basic reporting

This manuscript has been written in a professional way with accurate wording. Literature and background are sufficient for readers. It contains the required sections of a scientific paper, including article structures, figures, and raw data. No further edition is needed.

Experimental design

This study developed a heart failure risk prediction model for elderly patients after coronary rotational atherectomy (CRA) using machine learning algorithms. The retrospective cohort study involved 303 elderly patients, divided into heart failure and non-heart failure groups. The dataset was split randomly into a training set and validation set. Four algorithms (LR, XGBoost, SVM, LightGBM) were used to construct prediction models, with XGBoost showing the best performance. Key factors for heart failure included blood cholesterol, serum creatinine, fasting blood glucose, NT-proBNP (positive), and LVEF (negative). The XGBoost model demonstrated good predictive performance in the validation set. This model has the potential to enhance clinical decision-making and improve outcomes following CRA.

Validity of the findings

The XGBoost model demonstrated good predictive performance in the validation set. This model has the potential to enhance clinical decision-making and improve outcomes following coronary rotational atherectomy (CRA).

Reviewer 2 ·

Basic reporting

The manuscript tries to identify a risk prediction model for developing heart failure in elderly patients after coronary rotational atherectomy.
The structure is suitable and English is good. However the aims of the study are not very well described, The material and methods section of the manuscript needs more extensive description.

Experimental design

Methods should be described more thoroughly and the research question should be expressed in a more detailed manner.
I find that the two groups of patients (testing and training) are too heterogenous in the number of patients included. This also applies to the group with heart failure or without heart failure.

Validity of the findings

The conclusions are well stated, but why did the researchers focus only on the five criteria (that lack novelty) for predicting heart failure, as these anomalies are quite often in elderly patients?
Why is it important to assess the risk of heart failure only after coronary rotational atherectomy, and not after other percutaneous procedures? The procedure per se can modify the evaluated five criteria, so how do you distinguish the risk of heart failure form postprocedural complication?

Reviewer 3 ·

Basic reporting

The manuscript utilizes four ensemble machine learning algorithms to construct a postoperative heart failure risk prediction model for elderly patients undergoing Coronary Rotational Atherectomy (CRA). This model aims to assist clinical decision-making and improve adverse outcomes in post-CRA patients. The study holds significant value and implications. The manuscript presents a clear overall framework, effectively communicates its ideas.

Experimental design

The manuscript provides reliable experimental results.

Validity of the findings

The study findings are robust and effective.

Additional comments

Following the suggested revisions, it is suitable for publication:
1. The authors acknowledge the issue of sample imbalance in their research data. It is recommended to address this concern using appropriate techniques.
2. While performing the random data partitioning, it is important to clarify the methodology that was employed to ensure that all 53 heart failure patients were appropriately distributed across both the training and validation sets, avoiding concentration in a single dataset.
3. The authors are encouraged to provide an analysis of the factors that could potentially lead to an AUC of 1 for both the XGBoost and LightGBM models.
4. Including Calibration Curves is advised as an additional evaluation metric to assess the model's performance.

---

## Round 0.2 · accepted · Accept

All reviewers' comments have been adequately addressed.

Reviewer 1 ·

Basic reporting

The authors have addressed all of my concerns. I have no further questions.

Experimental design

The authors have addressed all of my concerns. I have no further questions.

Validity of the findings

The authors have addressed all of my concerns. I have no further questions.

Additional comments

The authors have addressed all of my concerns. I have no further questions.

Reviewer 3 ·

Basic reporting

The manuscript utilizes four ensemble machine learning algorithms to construct a postoperative heart failure risk prediction model for elderly patients undergoing Coronary Rotational Atherectomy (CRA). This model aims to assist clinical decision-making and improve adverse outcomes in post-CRA patients. The study holds significant value and implications. The manuscript presents a clear overall framework, effectively communicates its ideas.

Experimental design

The manuscript provides reliable experimental results.

Validity of the findings

The study findings are robust and effective.

Additional comments

The manuscript has been revised in accordance with the reviewer's suggestions and is approved for publication.